# Immune Recognition versus Immune Evasion Systems in Zika Virus Infection

**DOI:** 10.3390/biomedicines11020642

**Published:** 2023-02-20

**Authors:** Yee Teng Chan, Yi Ying Cheok, Heng Choon Cheong, Ting Fang Tang, Sofiah Sulaiman, Jamiyah Hassan, Chung Yeng Looi, Kim-Kee Tan, Sazaly AbuBakar, Won Fen Wong

**Affiliations:** 1Department of Medical Microbiology, Faculty of Medicine, University of Malaya, Kuala Lumpur 50603, Malaysia; 2Department of Obstetrics and Gynecology, Faculty of Medicine, University of Malaya, Kuala Lumpur 50603, Malaysia; 3School of Biosciences, Faculty of Health & Medical Sciences, Taylor’s University, 1, Jalan Taylors, Subang Jaya 47500, Malaysia; 4Tropical Infectious Diseases Research and Education Centre (TIDREC), Higher Education Center of Excellence (HICoE), University of Malaya, Kuala Lumpur 50603, Malaysia

**Keywords:** immune recognition, immune evasion, pathogen-recognition receptors, RIG-like receptor, type I interferon, Zika virus

## Abstract

The reemergence of the Zika virus (ZIKV) infection in recent years has posed a serious threat to global health. Despite being asymptomatic or mildly symptomatic in a majority of infected individuals, ZIKV infection can result in severe manifestations including neurological complications in adults and congenital abnormalities in newborns. In a human host, ZIKV is primarily recognized by RIG-like receptors and Toll-like receptors that elicit anti-viral immunity through the secretion of type I interferon (IFN) to limit viral survival, replication, and pathogenesis. Intriguingly, ZIKV evades its host immune system through various immune evasion strategies, including suppressing the innate immune receptors and signaling pathways, mutation of viral structural and non-structural proteins, RNA modulation, or alteration of cellular pathways. Here, we present an overview of ZIKV recognition by the host immune system and the evasion strategies employed by ZIKV. Characterization of the host–viral interaction and viral disease mechanism provide a platform for the rational design of novel prophylactic and therapeutic strategies against ZIKV infection.

## 1. Introduction

ZIKV belongs to the genus of *Flavivirus* in the Flaviviridae family that comprises multiple deadly human pathogens, including the dengue virus (DENV), Japanese encephalitis (JEV), the yellow fever virus (YFV), and the West Nile virus (WNV) [1]. First discovered in 1947 from *Macca malatta*, a Rhesus monkey in the Zika forest of Uganda [2,3,4], ZIKV has caused infection sporadically over the years, but infected individuals are typically asymptomatic or have mild symptoms such as a low-grade fever and a maculopapular rash. Following the unexpected 2015 to 2016 outbreak that occurred across 80 countries, particularly in Latin America [5,6,7], ZIKV has raised widespread concern and attracted interest from many researchers due to its clinical significance. This neurotropic virus preferentially targets human neural progenitor cells (NPCs) and causes sequelae in the immuno-privileged brain [8,9,10]. As a result, infection by ZIKV is associated with Guillain–Barré syndrome in adults [11,12,13,14] and congenital birth defects, including microcephaly and severe neurological defects, in children born to mothers infected during pregnancy [15,16,17,18,19]. There is currently no effective vaccine approved for ZIKV, although some are undergoing clinical trials [20,21]. Hence, further studies to characterize the host–pathogen interaction are imperative for minimizing the harm of a future disease outbreak. This review elaborates the virus properties, its recognition by human immune cells, and strategies utilized by the virus to evade its elimination by a human host.

## 2. ZIKV Properties

### 2.1. Gene and Structure

The ZIKV genome is made up of 10.8 kb positive-sense, single-stranded RNA flanked by the 5′ and 3′ untranslated regions (UTRs) with a single open reading frame (ORF) [22,23]. The ORF region encodes a single polypeptide, which is processed into three structural proteins, including a capsid (C), precursor membrane (prM), and envelope (E), as well as seven non-structural proteins (NS1, NS2A, NS2B, NS3, NS4A, NS4B, and NS5). The C proteins construct the icosahedral viral capsid, which encapsulates the viral genomic RNA, while the prM and E proteins are anchored on the outer membrane. prM is cleaved by host cell furin protease to generate mature virion, whereas the E protein is involved in binding and membrane fusion, which permit viral entry into the host cells [24].

NS1 is a glycoprotein of approximately 60 kDa that serves as an RNA replication complex in flaviviruses. Due to the importance of its replicative function in flaviviruses, the NS1 sequence is highly conserved, whereby ZIKV shares a >50% sequence homology with DENV2 and WNV NS1 [25]. The protein is found in the following different forms in various locations in the host cells: (i) dimers in membrane-bound vesicles in the lumen of the endoplasmic reticulum, (ii) dimers in association with the membranes of flavivirus-infected cells, and (iii) highly immunogenic hexamers that are secreted into extracellular fluid [26,27].

NS2A is a 22 kDa transmembrane protein located in the endoplasmic reticulum, and it plays a critical role in the viral replication process [28]. It also interacts with NS2B and NS3 to recruit viral RNA, prM, and E to the virion assembly site for virus morphogenesis [29,30]. It has been suggested that the NS2A protein participates in ZIKV-induced neurological damage, as it interacted with multiple adherent junctions in an embryonic mouse cortex and impaired radial glial cell proliferation in human forebrain organoids [31].

NS2B/NS3 forms the viral protease complex that is involved in genome replication and cleavage of the viral polypeptide [32]. NS3 carries the protease domain at the N-terminus and the RNA helicase domain at the C-terminus, while NS2B acts as the membrane-bound domain that positions NS3 to its substrate and forms part of the NS3 catalytic domain for substrate binding [33,34].

NS4A/NS4B cause neurological impairment via manipulating the cellular survival and autophagy signaling pathways [35]. The introduction of NS4A or NS4B in human fetal neural stem cells (NSCs) resulted in impaired neurosphere formation, likely through inhibiting Akt kinase phosphorylation at Thr308 and Ser473 and mammalian target of rapamycin (mTOR) phosphorylation at Ser2448, which disrupted autophagy [36]. NS5, on the other hand, comprises methyltransferase with a short linker to the RNA-dependent RNA polymerase (RdRP) that is vital for RNA replication. It performs guanylyl transferase activity to catalyze the de novo formation of a methylated RNA cap structure using a triphosphorylated RNA transcript [37].

### 2.2. African and Asian ZIKV Lineages

Phylogenetic analysis has classified the ZIKV into two major genotypes, namely, the African and Asian lineages; the latter is further subdivided into the local Asian or contemporary American subclades [38,39]. The African and Asian ZIKV lineages display differences in virulence, transmissibility, and replication kinetics [40,41,42], despite sharing a high degree of similarity (>88.9%) in their genomic sequences [22]. The African ZIKV strain demonstrates a higher rate of transmissibility in the mosquito vector *Aedes aegypti* compared to its Asian counterpart [43]. Its infection results in a higher rate of lethality and can lead to cases of fetal death [44]. In contrast, the low-virulence Asian lineage does not induce early cell death, but it may lead to chronic infections in the fetal central nervous system [45]. The reemergence of ZIKV epidemics in 2015 were dominated by a strain of Asian ZIKV lineage that is commonly named the American strain [46]. Preceding the outbreak, ZIKV Asian lineage had been associated with an evolutionary mutation in the viral E gene (V473M) during replication and transmission between mosquito and host [47]. This mutation increases its virulence and viremia generation, hence enhancing transmission, which could be a critical determinant in the epidemics. Intriguingly, an effort to inverse the V473M substitution in the epidemic ZIKV strain isolated in Puerto Rico in 2015 reversed the pathogenic phenotypes of the virus [47]. Recent ZIKV outbreaks of the local Asian lineage have been reported in different states of India in 2018 and 2021 [48,49,50].

### 2.3. Transmission and Life Cycle

Similar to other flaviviruses, ZIKV is vector-borne and can be disseminated by infected female *Ae. aegypti* and *Ae. albopictus* mosquitoes. However, it differs from DENV in that it can be transmitted vertically from a pregnant mother to a baby [51,52], via blood transfusions [53], and via sexual intercourse [54,55]. Vertical transmission is observed in the mosquito vectors, *Ae. aegypti* and *Ae. albopictus,* to the larvae of infected mosquitoes [51,56].

The life cycle of ZIKV is highly similar to other members of the Flavivirus family; it begins with the entry of a viral particle into a host cell via clathrin-mediated endocytosis modulated by the binding of viral protein E. Viral entry is facilitated by the rolling and accumulation of viral particles along a host cell surface. The differential expression of various binding factors in a host cell surface dictates the viral tropism. The presence of the transmembrane receptor tyrosine kinase protein anexelekto (AXL), which is highly expressed by neural cells, dendritic cell-specific intracellular adhesion molecule 3-grabbing nonintegrin (DC-SIGN), tyrosine-protein kinase receptor (TYRO3,) and T-cell immunoglobulin and mucin domain 1 (TIM-1) on host cells is vital for the viral endocytic event [57,58].

When ZIKV reaches a clathrin-expressing surface, the host cell membrane invaginates and fuses with the viral membrane in the presence of acidic host cell cytoplasm, allowing the viral genome to be released into the cytoplasm (Figure 1). Following the release, protein translation occurs, and the newly synthesized viral proteins will be recruited into the endoplasmic reticulum for assembly [59]. With help from the NS proteins, new and immature viral particles migrate to the Golgi body, where precursor prM proteins are cleaved. Mature virions are subsequently released from the cell and are ready for a new cycle of infection. Occasionally, an immature viral particle carrying the uncleaved prM can be released [60].

### 2.4. Symptoms Caused by ZIKV Infection

During the Yap Island outbreak in 2007, a majority of the cases were mild, with clinical symptoms that included low-grade fever, maculopapular rash, arthralgia, and conjunctivitis [61]. Severe neurological complications of ZIKV infection were observed in a small number of cases during the French Polynesian outbreak. This was highlighted by the increased prevalence of an autoimmune disease causing acute or subacute flaccid paralysis, known as Guillain–Barre syndrome, to approximately a 20-fold higher rate than was expected (1/2 in 100,000 people per year) in adults, approximately 3 weeks following the ZIKV outbreak [62]. Trends of microcephaly among newborns of infected mothers were reported during the outbreak in Brazil from 2015 to 2016 [63]. Other forms of neurological deficits, including meningoencephalitis [64,65], transverse myelitis [66], ophthalmic manifestation with optic nerve and retina complications [67,68], and other neuronal developmental defects [69], were identified among infants. Subsequent studies using human brain organoids [70], as well as animal models using macaques, mice, or fruit flies [52,71,72], have confirmed the viral neurotropism and developmental impact. Early neurological impairments, including severe intellectual disability, spastic tetraparesis, dysphagia, and failure to thrive [73], as well as severe motor impairment, were recently described in congenital ZIKV-infected children [74].

ZIKV causes neurological deficits through damaging neuronal development and proliferation. Li, et al. [15] showed that in human NPCs, ZIKV infection caused cell-cycle arrest, apoptosis, and the inhibition of cell differentiation, which eventually gave rise to cortical thinning and microcephaly. Gabriel, et al. [10] reported that ZIKV infection resulted in the premature differentiation of NPCs, which was associated with centrosome perturbation, progenitor depletion, disrupted ventricular zone proliferation, impaired neurogenesis, and cortical thinning. In addition, Onorati, et al. [75] utilized a single-cell RNA-sequencing technique to investigate the effects of ZIKV on the neuropathogenesis of neocortical and spinal cord neuroepithelial stem cells, and they demonstrated that ZIKV infection caused disrupted cell mitoses, supernumerary centrosomes, structural disorganization, and cell death. Treatment with nucleoside analogs inhibited ZIKV replication and ZIKV-mediated death in neuroepithelial stem cells [75].

## 3. Innate Immune Recognition

Following the viral invasion of host cells, ZIKV viral components, including nucleic acids and proteins, are recognized as foreign substances or pathogen-associated molecular patterns (PAMPs) by the host innate pathogen-recognition receptors (PRRs) [76]. Upon recognition, these receptors transduce signals into the nucleus to initiate a robust immune response to eliminate the virus. Toll-like receptors (TLRs), particularly TLR3, TLR7, and TLR8, as well as retinoic acid-inducible gene I (RIG-I)-like receptors (RLR), including RIG-I and the melanoma differentiation-associated gene 5 (MDA5) receptors, are the host innate receptors located in either the endoplasmic vesicles or cytoplasm. These receptors play crucial roles in triggering signaling pathways and initiating an antiviral response in the host [77].

### 3.1. ZIKV Recognition by RIG-I Receptor

Principally, the ZIKV RNA is recognized by the cytosolic RNA helicase, known as RIG-I and MDA5, at the early phase of infection (Figure 2) [78,79,80]. Human iPSC-derived astrocytes sense ZIKV with both RIG-I and MDA5 to mount a strong antiviral cytokine response that includes the secretion of type I interferon (IFN-α and IFN-β) and pro-inflammatory cytokines such as interleukin-6 (IL-6) [81]. The RIG-I receptor binds to the highly structured and conserved 5′ region of the newly synthesized ZIKV transcripts before the capping process takes place [82]. At a later stage, MDA5 functions as a secondary PRR that binds the long viral RNA and augments the innate antiviral response initiated by RIG-I [83]. Upon detecting the cytoplasmic viral RNA, RIG-I alters its conformation, exposing the caspase activation and recruitment domain (CARD) which interacts with the mitochondrial antiviral signaling proteins (MAVS) [78,84,85,86]. The MDA5 double-stranded RNA (dsRNA) dimer polymerizes and induces the aggregation of MAVS, resulting in signal amplification [87]. This initiates a signaling cascade, leading to the expression of type I IFNs and IFN-stimulated genes (ISGs), as elaborated below.

### 3.2. ZIKV Recognition by TLR3

A recent study has reported a significant increase in TLR3 expression when cells are infected with ZIKV [88]. Viral RNA recognition by TLR3 enhances the production of inflammatory cytokines, including IL-6 [81]. TLR3 also suppresses the type I IFN response triggered by RIG-I in a suppressor of cytokine signaling 3 (SOCS3)-dependent manner. The pharmacological inhibition or genetic disruption of TLR3 in astrocytes caused a decrease in viral titers and in the viral-induced inflammatory response in infected astrocytes, and partially restored the deficits caused by ZIKV infection [81,89].

Using human embryonic stem-cell-derived cerebral organoids and mouse neurospheres, Dang, et al. [90] showed that ZIKV infection upregulated TLR3 and caused diminished organoid volume that was reminiscent of microcephaly. In contrast, a TLR3 blockade reduced the phenotypic effects of ZIKV infection. Therefore, ZIKV-mediated TLR3 activation likely participates in the mechanistic control of the neurogenesis disruption that leads to serious neurological disorders, including microcephaly, in newborns [90].

Importantly, a vital role of TLR3 in ZIKV infection has not only been shown in experimental animal models, but it has also been supported by clinical findings. Clinical data using ZIKV-infected patient samples have suggested a significant upregulation of TLR3 mRNA transcript in patients, and its expression level was correlated with the expression of cytokines such as IL-12, tumor necrosis factor-α (TNF-α), and interferons (IFN-α, -β and -γ) [88]. Furthermore, the TLR3 gene rs3775291 single-nucleotide polymorphism (SNP) was associated with the occurrence of a cluster of malformations, which was named congenital Zika syndrome (CZS) [91]. This missense SNP in TLR3 caused the decreased binding capacity of dsRNA, resulting in impaired antiviral activity and an increased ZIKV viral load [92].

### 3.3. ZIKV Recognition by TLR7/8

In addition to the TLR3 recognition of dsRNA, other endosomal receptors such as TLR7 and TLR8 recognize ZIKV single-stranded RNA (ssRNA). Vanwalscappel, et al. [93] investigated the involvement of TLRs in ZIKV infection by treating monocytes and macrophages with different TLR agonists. Among the different agonists tested, the TLR7/8 agonist R848 (resiquimod) demonstrated the most potent inhibitory effect on ZIKV replication [93]. TLR7/8 agonists induced the expression of various genes, including viperin, an interferon-induced gene. The gene-editing-mediated deletion of viperin in macrophages facilitated viral growth in the host cells, whereas a lentiviral-mediated transduction of viperin in microglial CHME3 cells rendered resistance to viral replication. As such, TLR agonists have been suggested to be a potential prophylactic or therapeutic treatment option for ZIKV [93]. Nevertheless, it must be noted that the clinical data show that, in contrast to TLR3 mRNA, there was no increase in TLR7 or TLR8 mRNA levels detected in the ZIKV-infected patients [88]. Hence, more studies should be conducted to inspect the role of TLR7 and TLR8 in ZIKV infection using, for example, a knockout mouse system.

### 3.4. Signaling Pathway Activated by ZIKV Recognition

TLRs activate the myeloid differentiation primary response (MyD88), and they transmit signals via signaling molecules such as TNF receptor-associated factor 3 (TRAF3) and TRAF6 [94,95]. Both RIG-I and MDA5 migrate to the mitochondria and stimulate the MAVS signaling cascade. These signaling pathways subsequently activate either the inhibitor of nuclear factor kappa-B (NF-κB) kinase subunit (IKKα/β) or the IKKε and TANK binding kinase 1 (TBK1) [96,97], which eventually results in the activation of the transcription factors NF-κB, IRF-3, and IRF-7 [98]. These events eventually initiate the expression of type I IFN for antiviral defense [99].

The binding of type I IFN to its receptor induces the transcription of IFN-stimulated genes (ISGs) that suppress viral infection through the Janus kinase/signal transducer and activator of transcription (JAK/STAT) pathway [86,95,100]. The release of IFN by ZIKV-infected cells establishes an antiviral state which stimulates cells in an autocrine or a paracrine manner to upregulate the expression of RIG-I and MDA5 genes [101,102]. This generates a positive feedback mechanism, resulting in a higher IFN production rate and the expression of ISGs to rapidly build up a vigorous antiviral response (Figure 2). Following the binding of type I IFN to its receptor, JAK1 activation leads to STAT2 phosphorylation and the formation of the interferon-stimulated gene factor 3 (ISGF3) complex, which consists of the STAT1, STAT2, and IRF-9 triad. The ISGF3 is then translocated into the nucleus and transcribes ISGs, as well as the SOCS, which negatively regulate the JAK/STAT pathway by ubiquitinating JAK1 and promoting degradation through the proteosome [95]. Interestingly, ZIKV NS4A activates the ISGF3 signaling pathway and induces the upregulation of ISGs to restrict viral replication while the NS4A blockage removes the antiviral effect [103], which, presumably, acts as negative feedback to maintain viral persistence.

Mice deficient in type I IFN signaling exhibit severe pathology and succumb to ZIKV infection [104,105]. Because of the ZIKV infection interferon receptor (*Ifnar1*-/-) or *Irf3*-/-*Irf5*-/-*Irf7*-/-, triple-knockout mice developed neurological disease and sustained high viral loads in the brain, spinal cord, and testes [105]. AG129 mice, with deficient IFN-α, -β, and -γ receptors, were highly susceptible to ZIKV infection, and they demonstrated rapid viremic dissemination to visceral organs and the brain and succumbed at approximately one week post-infection [104]. Using an anti-IFNAR1-treated, Rag1-/- mouse model, vertical ZIKV transmission in postnatal mice resulted in structural abnormalities and increased cell death in multiple regions of the brain [106]. These data collectively highlight a crucial role of the IFN signaling pathway in providing protection against ZIKV infection.

### 3.5. Low Pattern-Recognition Receptors in the Lower Female Reproductive Tract Enables Viral Replication

A recent study utilized macaque and mouse models to examine PRRs in tissues derived from uninfected subjects and subjects that were vaginally infected with ZIKV. It was shown that the basal expression levels of RNA-sensing PRRs are scarce in the lower female reproductive tract, and vaginal ZIKV infection minimally stimulates PRR expression [107]. Consequently, ZIKV recognition by PRRs in the lower female reproductive tract provided limited protection to the host against viral replication, and this rendered a high viral load following infection. Nevertheless, it was demonstrated that PRRs are required to prevent further dissemination of ZIKV to the upper female reproductive tract or to other tissues [107]. This further supports the importance of PRR-mediated innate immunity in dampening viral replication and systemic dissemination in the host.

## 4. ZIKV Attenuates Innate Recognition

### 4.1. ZIKV Modulates the Translocation of RIG-I and MDA5

Previous studies have reported that ZIKV NS proteins antagonize infection-mediated type I IFN production through various mechanisms to benefit viral replication in the host cells (Figure 2). Using the ZIKV NS4A-overexpression system, Hu, et al. [108] demonstrated that ZIKV NS4A binds directly to MAVS to interrupt its interaction with RIG-I through the CARD and transmembrane domains. This binding disrupts MAVS localization from the cytoplasm to the mitochondria and results in the diminished production of type 1 IFN [108].

Furthermore, Riedl, et al. [109] reported that ZIKV NS3 mimics the binding motif of the 14-3-3 molecule. The 14-3-3 family includes two members, i.e., 14-3-3ε and 14-3-3η, which promote the cytosolic-to-mitochondrial translocation of RIG-I and MDA5, respectively [109]. The binding of ZIKV NS3 to 14-3-3 inhibits the interaction of 14-3-3 with RIG-I and MDA5, preventing their localization to the mitochondria and resulting in attenuated antiviral signaling [109].

### 4.2. ZIKV Degrades the cGAS/STING Pathway

GMP-AMP synthase (cGAS) is one of the key recognition receptors that functions as a DNA sensor in the cytosol. cGAS activates the stimulator of interferon genes (STING) and triggers the activation of innate immunity through the TBK1 kinase. Although cGAS is a DNA sensor, recent studies have suggested its involvement in a host immunity to ZIKV infection. In the genetically tractable *Drosophila* system, it has been shown that STING restricts ZIKV infection by inducing autophagy in the brain [71].

Intriguingly, several studies have shown the ability of ZIKV to avoid STING-mediated protection in the host cells. Zheng, et al. [110] reported that ZIKV NS1 evades immune sensing through caspase-1-mediated cGAS degradation. On the other hand, Ding, et al. [111] demonstrated that ZIKV NS2B-NS3 cleaves and degrades the cGAS/STING pathway in non-human primate cells. Notably, the presence of a protease cleavage site at position 78/79 of the human STING molecule and its absence in a rodent counterpart have been shown to render viral tropism, specifically in humans and primates [111].

### 4.3. ZIKV Blocks TBK1 Phosphorylation

The ZIKV NS1 protein hampers IFN-β production in dendritic cells via binding to TBK1 [86,112]. NS1 binding inhibits the TBK1 kinase phosphorylation of IRF-3 and IRF-7 transcription factors, and, thus, it impairs the downstream transactivation activity of type I IFN [86,112].

Aside from NS1, the ZIKV NS2A, NS2B, and NS4B proteins participate in the inhibition of TBK1 phosphorylation, whereas ZIKV NS4A impairs IRF-3 phosphorylation, thus suppressing type I IFN production [112]. In human neuroepithelial stem cells, ZIKV infection disrupts the localization and activity of TBK1 by sequestering the phosphorylated TBK1 to the mitochondria during mitosis [75].

### 4.4. ZIKV Represses the Promoter Activity of the NF-κB and IRF-3 Transcription Factors

The main transcription factors in innate immunity to ZIKV infection include NF-κB, IRF-3, and IRF-7. Some reports have suggested that the transactivation activity of these transcription factors can be suppressed by different ZIKV molecules. A Luciferase assay in HEK293T showed that ZIKV NS2A and NS4A dramatically repress the NF-κB promoter activity induced by the MDA5/RIG-I signaling pathway [113]. In addition, ZIKV NS5 interacted with and suppressed IRF3 to prevent its transcriptional activity in the host cells [112].

### 4.5. ZIKV Disrupts the JAK/STAT Signaling Pathway

In the JAK/STAT signaling pathway, ZIKV NS2B-NS3 inhibits virus-induced apoptosis and depletes JAK1 to prevent the induction of antiviral ISGs [114]. The inhibition of JAK1 using ruxolitinib significantly increased ZIKV replication in human Hoffbauer cells, trophoblasts, and neuroblasts [115]. ZIKV NS2A promoted the degradation of STAT1 and STAT2, which impeded the JAK/STAT cascade [116]. In addition, the ZIKV NS4B and NS5-noncoding RNA interaction suppressed STAT1 phosphorylation and blocked the nuclear localization of STAT1 and STAT2, resulting in impaired IFN signaling [117,118].

The ZIKV NS5 protein, on the other hand, promoted the proteasomal degradation of STAT2 [119,120]. An experiment using a STAT2-deficient mouse model demonstrated a high susceptibility to ZIKV infection and viral dissemination to the central nervous system, gonads, and other visceral organs, and it displayed neurological symptoms [121]. This suggests the role of STAT2 in limiting ZIKV replication and pathology, and its degradation by ZIKV molecules could result in severe pathology in a human host.

Additionally, AXL, which is expressed in human glial cells and astrocytes, also mediates ZIKV infection by dampening type I IFN signaling [122]. AXL attenuates the ZIKV-induced type I IFN signaling genes through modulating SOCS1, a type I IFN signaling suppressor, in a STAT1/STAT2-dependent manner.

### 4.6. ZIKV Suppresses Type I IFN Signaling through Inducing Inflammasome Activity

ZIKV infection causes severe inflammation through NOD-, LRR-, and pyrin domain-containing protein 3 (NLRP3) inflammasome-mediated IL-1β production [110,123,124]. ZIKV NS5 facilitates the assembly of the NLRP3 inflammasome complex through binding to NLRP3 [123,124]. In addition, ZIKV NS1 inhibited the proteasomal degradation of caspase 1 by recruiting host deubiquitinase, a ubiquitin-specific peptidase 8 (USP8), to cleave the poly-ubiquitin chains [110]. Cells or mice deficient in NLRP3 exhibited a decreased secretion of IL-1β and increased type I IFN production following ZIKV infection, as well as increased host resistance to ZIKV-induced effects in vivo and in vitro [110,124]. Together, these findings suggest that through enhancing inflammasome activity, ZIKV antagonizes type I IFN signaling to benefit its replication in host cells.

### 4.7. ZIKV Antagonizes RNAi-Mediated Antiviral Activity

RNA interference (RNAi), a posttranscriptional gene-silencing mechanism, can act as an intrinsic antiviral mechanism [125]. In the process of antiflaviviral RNAi, the host endoribonuclease Dicer recognizes and cleaves the viral dsRNA replicative intermediates into virus-derived small interfering RNAs (vsiRNAs). Subsequently, the Argonaute protein (AGO) of the RNA-induced silencing complex (RISC) utilizes vsiRNAs to destruct viral RNAs in the infected cells [126].

To evade immune response, ZIKV encodes viral suppressors of RNAi (VSR) to antagonize RNAi-mediated antiviral immunity [127]. The VSR activity of ZIKV NS2A suppressed antiviral RNAi in vitro through the inhibition of vsiRNA production [128]. ZIKV C protein is also a VSR that directly interacts with and antagonizes the endoribonuclease activity of host Dicers in human NSCs [129,130]. The vsiRNA production is Dicer-dependent, as evidenced by knockdown of Dicer in the RNAi pathway resulting in reduced vsiRNA and enhanced ZIKV replication in NSCs [130]. Furthermore, Enoxacin, an RNAi enhancer, has been shown to inhibit ZIKV-induced phenotypes associated with microcephaly by increasing RNAi in brain organoids [131], emphasizing the significant role of antiviral RNAi against ZIKV infection.

In the ZIKV-infected mouse embryonic brain, capsid-mediated Dicer inhibition disrupts the production of host microRNAs (miRNA) that are essential for neural development (i.e., let-7a, miR-9, miR-17, and miR-19a), thus causing severe defects in neurogenesis in vitro and corticogenesis in utero [129]. In contrast, using a capsid-H41R mutant, ZIKV reduces its pathogenicity to cause neurologic deficits, which is due to the loss of capsid–Dicer interaction and failure to inhibit miRNA production. Accordingly, ZIKV dysregulates the miRNA–mRNA interaction network that negatively impacts several biological processes such as the cell cycle and neurogenesis in human NSCs [132], as well as in fetal astrocyte SVG-A cells [133]. Remarkably, the modulation of RNAi activity through hijacking host Dicers, vsiRNA, and miRNA productions represents a refined mechanism for ZIKV immune evasion and favors its replication.

## 5. Other Evasion Strategies Exploited by ZIKV

### 5.1. ZIKV Evades Immune Attack through Gene Mutation

ZIKV evades host immune response by exploiting various strategies, including genetic adaptation, perturbation of the IFN signaling and complement pathways, mimicking the host RNA structure, and modulating humoral immunity [38,114].

A single mutation (serine-to-asparagine substitution) at residue 139 of ZIKV prM causes the virus to become more infectious and lethal [134]. An in vitro study demonstrated that the ZIKV VEN/2016 strain induced viral replication, severe neuropathology, and higher mortality rates in human and mice NPCs [134]. Further, isoleucine-to-valine (I39V) or isoleucine-to-threonine (I39T) mutations at the amino acid residue 39 of the NS2B increase ZIKV replication and transmission in human NPCs [135]. A188V mutation of the ZIKV NS1 gene promotes the high antigenic effect of the NS1 protein and enhances viral infectivity [136].

Furthermore, the highly conserved N-linked glycosylation in the amino acid of the E protein in the ZIKV Asian and American strains has been shown to mediate neurotropism and cause neurological damage [137]. This single glycosylation site modulates the ZIKV interaction with the neutralizing antibodies and receptors [138], and, hence, it provides ZIKV with the ability to survive and multiply in a human host.

### 5.2. ZIKV Alters Cellular Processes

ZIKV infection causes swollen mitochondria in the human neurosphere [70]. Mitochondrial fragmentation and disrupted mitochondrial membrane potential following ZIKV infection has been reported in human NSCs and in a glioblastoma cell line [139].In addition, the ZIKV NS4B protein induces mitochondrial elongation during ZIKV infection. Mitochondria elongation also occurs following DENV infection, and it generates a favorable condition for viral replication [140].

The ZIKV NS4A and NS4B proteins are related to microcephaly due to their perturbation of neurogenesis through dysregulating the Akt-mTOR pathway and autophagy in fetal NPCs [36]. Failure to inhibit autophagy facilitates ZIKV propagation and pathogenesis. The NS1 protein interrupts the complement pathway by blocking the polymerization of complement component C9 and the membrane attack complex formation [141].

Moreover, perturbation of natural killer cell-mediated lysis during ZIKV infection has been demonstrated in an in vitro study, with clear evidence of the major histocompatibility complex (MHC) class I being upregulated on the surface of infected cells to antagonize the cell lysis by natural killer cells [142].

### 5.3. ZIKV Forms an RNA Cap through Methyltransferase Activity

Under common circumstances, viral mRNA that lacks 2′-O-methylation at 5′ cap is detected by the IFN-inducible protein with tetratricopeptide (IFIT) to restrict viral propagation [143]. The NS5 protein executes methyltransferase activity to form a 2′-O-methylated RNA cap, which mimics the host RNA cap structures; this reaction, which aids viral escape from IFTI recognition, has also been reported in other flaviviruses, including WNV, DENV, and JEV [143,144,145]. A comparative study of ZIKV NS5 methyltransferase revealed that ZIKV possesses the same features for avoiding a host immune response [146]. Furthermore, the incomplete degradation product of ZIKV RNA, also known as sub-genomic flaviviral RNA (sfRNA), interacts with and depletes the inhibition of viral translation mediated by the fragile X mental retardation protein (FMRP) [147].

### 5.4. ZIKV-Mediated Modulation of Humoral Immune Response

ZIKV has evolved unique approaches to modulate the host humoral immune response. For instance, ZIKV RdRP has low fidelity, which allows the rapid development of antigenic variation in the epitopes of the ZIKV E protein DIII. This drives antigenic escape and prevents epitope recognition by host-specific antibodies and T cell receptors [148]. Genome-wide transposon mutagenesis screening has disclosed the ZIKV ability to tolerate mutation, particularly in structural proteins such as E protein, which are more permissive to genetic modifications [149]. This genetic flexibility acquired by ZIKV greatly impacts the ability of human adaptive immunity to rapidly neutralize and eliminate the pathogen.

Notably, the molecular interactions between ZIKV and its host are intricate, as various viral–host factors are involved in establishing infection and suppressing viral load. Nevertheless, ZIKV has exploited several advanced survival tactics in hosts to sustain infection. A summary of ZIKV-mediated evasion mechanisms is given in Table 1 below.

## 6. Conclusions

The reemergence of ZIKV in recent years has resulted in public health emergencies worldwide. Host innate immunity is essential for controlling virus infection and eliminating the virus. The recognition of ZIKV by innate immune receptors, such as RIG-I, initiates signaling pathways and activates the host defense system, primarily through the secretion of type I IFN to program an antiviral state in infected or neighboring cells. Through employing various immune evasion approaches, ZIKV promotes infection, replication, and dissemination, and it is detrimental to the brain and nervous system in adults and fetuses. Therefore, although most studies have focused on antagonizing the ZIKV NS proteins, studies on host aspects are equally crucial for providing insights for the rational design of therapeutic drugs and vaccines for ZIKV.

## Figures and Tables

**Figure 1 biomedicines-11-00642-f001:**
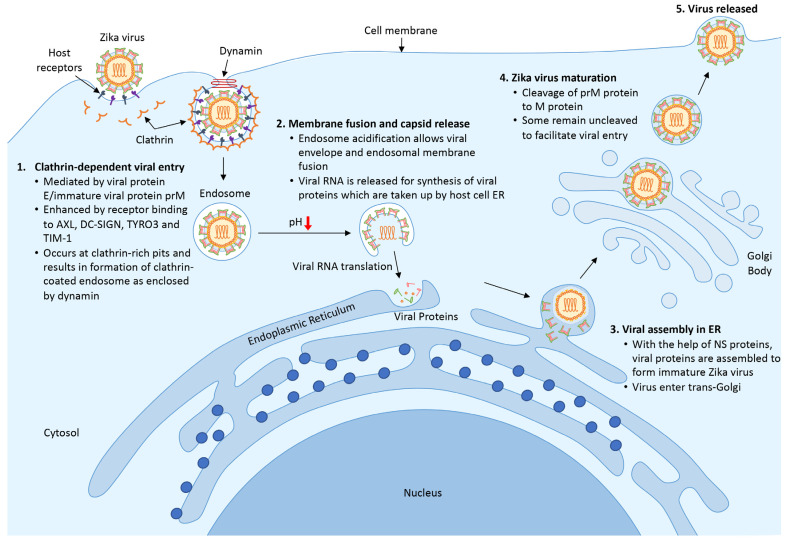
Life cycle of ZIKV. (1) ZIKV encounters host cells and binds to host receptors (AXL, DC-SIGN, TYRO3, and TIM-1) via viral protein E and prM and initiates clathrin-dependent viral entry. (2) Upon entering the cell, the endosome matures and acidifies, resulting in the release of viral RNA and the translational process to synthesize viral proteins. (3) New viral proteins are assembled into an immature viral particle within the endoplasmic reticulum. (4) Immature viral particles enter the trans-Golgi network where prM protein is cleaved into a mature virus. Finally, the newly formed virus is released to the surrounding areas and is ready for subsequent infection.

**Figure 2 biomedicines-11-00642-f002:**
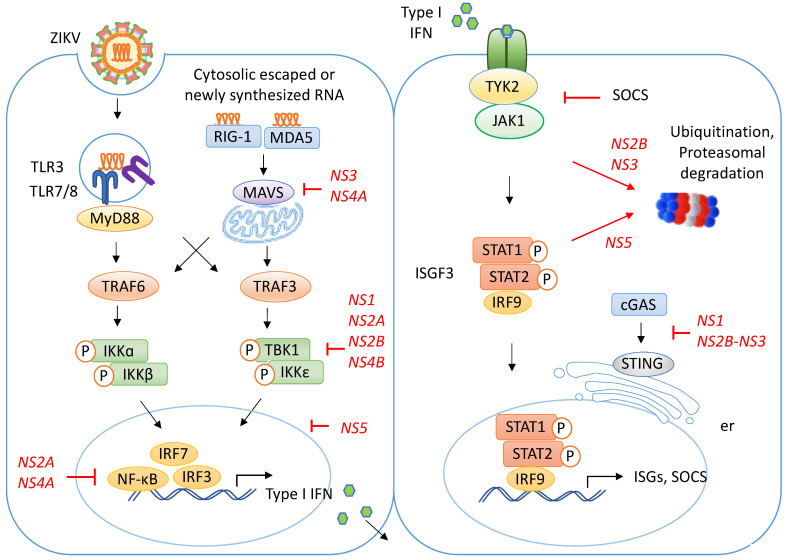
Immune recognition pathway in ZIKV-infected cells. The endosomal TLRs (TLR3, TLR7, and TLR8) and the cytosol RIG-like receptors (RIG-I and MDA5) recognize viral invasion through binding to ZIKV RNA. The TLRs activate MyD88, whereas RIG-I and MDA5 initiate the MAVS signaling cascade. These result in the activation of TRAF3 or TRAF6. TRAF6 subsequently activates IKKα and IKKβ, which result in NF-κB nuclear translocation and the transactivation of various immune-response-associated genes. On the other hand, TRAF3 causes TBK1 and IKKε phosphorylation, which, in turn, activates IRF-3 and IRF-7 for the transactivation of type I IFN (IFNα and IFNβ). Type I IFN cytokines are released to stimulate antiviral defense in either an autocrine or a paracrine manner. The binding of secreted-type I IFN to its receptor leads to JAK1 activation, STAT2 phosphorylation, and the formation of the ISGF3 complex that is translocated into the nucleus to transcribe ISGs. The viral particles NS3 and NS4A inhibit MAVS migration to the mitochondria. NS1, NS2A, NS2B, and NS4B prevent the phosphorylation of TBK1. NS2A and NS4A suppress NF-κB activity, while NS5 binds to IRF-3 and inhibits its transactivation activity. Furthermore, NS2B and NS3 suppress JAK1 signal transduction by inducing SOCS-dependent ubiquitination and degradation through the proteosome, whereas NS5 modulates the proteasomal degradation of STAT2, and, hence, the downstream signaling is prevented, repressing ISGs production. NS1 and NS2B-NS3 interrupt the cGAS/STING pathway to weaken the host immunity to ZIKV. Mitochondrial, mt; endoplasmic reticulum, er.

**Table 1 biomedicines-11-00642-t001:** Summary of the immune evasion mechanisms of ZIKV.

Host Immune Pathways	ZIKV Protein	Mechanism	References
RIG-I and MDA5 signaling	NS4A	Binds to MAVS and interrupts RIG-I interaction Disrupts MAVS localization to the mitochondria	[108]
	NS3	Binds to 14-3-3 molecule and inhibits its interaction with RIG-I and MDA5 Prevents RIG-1 and MDA5 localization to the mitochondria	[109]
cGAS/STING pathway	NS1	Promotes caspase-1-mediated cGAS degradation	[107]
	NS2B-NS3	Cleaves and degrades the cGAS/STING pathway	[112]
TBK1 phosphorylation	NS1	Inhibits the TBK1 phosphorylation of IRF-3 and IRF-7	[86,112]
	NS2A, NS2B, and NS4B	Inhibits TBK1 phosphorylation	[112]
	NS4A	Impairs IRF-3 phosphorylation	[112]
NF-κB and IRF-3	NS2A and NS4A	Represses NF-κB promoter activity	[113]
	NS5	Suppresses IRF-3 transcriptional activity	[112]
JAK/STAT pathway	NS2B-NS3	Inhibits virus-induced apoptosis and depletes JAK1	[114]
	NS5	Promotes proteasomal STAT2 degradation	[119,120]
	NS2A	Degrades STAT1 and STAT2	[116]
	NS4B	Suppresses STAT1 phosphorylation and blocks the nuclear localization of STAT1 and STAT2	[117,118]
	NS5-noncoding RNA	Suppresses STAT1 phosphorylation and blocks the nuclear localization of STAT1	[117,118]
Type I IFN signaling	NS1	Inhibits the proteasomal degradation of caspase 1 that cleaves cGAS	[110]
	NS5	Binds to NLRP3 and induces inflammasome-mediated IL-1β production	[123,124]
Antiviral RNAi	NS2A	Inhibits vsiRNA production	[128]
	C	Interacts with and antagonizes the endoribonuclease activity of Dicers	[129,130]
Gene mutation	pRM	Mutation at residue 139	[134]
	NS2B	Mutation at residue 39	[135]
	NS1	Mutation at residue 188	[136]
	E	N-linked glycosylation in amino acid mediates neurotropism and neurological damage This glycosylation modulates ZIKV interaction with neutralizing antibodies and receptors	[137]
Cellular processes	NS4B	Mitochondrial elongation	[140]
	-	Swollen mitochondria Mitochondrial fragmentation and disrupted mitochondrial membrane potential	[70] [139]
	NS4A and NS4B	Modulates Akt-mTOR signaling and autophagy, hence perturbs neurogenesis	[36]
	NS1	Interrupts complement pathway	[141]
	-	Antagonizes natural killer cell-mediated lysis	[142]
Methyltransferase activity	NS5	Forms 2’-O-methylated RNA cap to mimic host RNA cap	[146]
Humoral immunity	RdRP	Develops antigenic variation in the E protein DIII epitopes	[148]
	E	Permissive to genetic modifications	[149]

## Data Availability

Not applicable.

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
