# Peer review of "Immune Recognition versus Immune Evasion Systems in Zika Virus Infection"

_biomedicines, 2023, doi:10.3390/biomedicines11020642_

Round 1

Reviewer 1 Report

This review by Yee Teng Chan et al. describes the Zika virus infection with emphasis on immune recognition and evasion strategies exploited by the virus. It’s a well-written and useful summary of the ZIKV disease mechanism. My comments are below:

1)    A tabulated summary of Immune evasion strategies in ZIKV infection could be a useful addition to the review. Authors should consider incorporating a table focusing on evasion strategies by ZIKV.

2)    In section 2.2, the authors mention the dominant strain in the recent 2015 epidemic, more information is required. 

3)    Authors should also include the immune evasion mechanism of ZIKV capsid interaction with endoribonuclease in the immune-privileged organs in a separate section.

4)    Sections 3.1 and 3.2 seem incomplete. The authors should add a few more lines to conclude. Section 3.5 needs to be revised with more supporting information.

5)    In the review, some sentences need rephrasing for clarity, for instance, lines 103-105, 188-190, and 209-210.

6)    Abbreviations should be used all throughout the review once the expanded full form has been mentioned at the beginning. The authors should revise thoroughly.

7)    References missing from lines 117-124, 169-170, 224-226, 241-244. Doi missing from references 2, 52, and 61.

8)    Typo error- lines 84-85, 99, 378.

Author Response

We sincerely thank the reviewer for valuable comments provided to improve our manuscript. Our response to the comments are as follow:

  1. We have tabulated a summary of immune evasion strategies in ZIKV infection, as suggested by the reviewer (Table 1).

  1. We have added some information to the section 2.2, as suggested (Page 3 Line 102-110).

Section 2.2 : "The reemergence of ZIKV epidemics in 2015 is dominated by a strain of Asian ZIKV lineage that is commonly named as the American strain [47]. Preceding the outbreak, ZIKV Asian lineage has been associated with an evolutionary mutation in the viral E gene (V473M) during replication and transmission between mosquito and host [48]. This mutation increases its virulence and viremia generation, hence enhances transmission, which could be a critical determinant in the epidemics. Intriguingly, an effort to inverse the V473M substitution in the epidemic ZIKV strain isolated in Puerto Rico in 2015 re-verses the pathogenic phenotypes of the virus [48]. Recent ZIKV outbreaks of the local Asian lineage have been reported in different states of India in 2018 and 2021 [49-51]."

  1. We have include the information on the immune evasion mechanism of ZIKV capsid interaction with endoribonuclease to the new section 4.7, as suggested (Page Line).

Section 4.7 : "RNA interference (RNAi), a posttranscriptional gene-silencing mechanism, can act as an intrinsic antiviral mechanism [128]. In the process of antiflaviviral RNAi, the host endoribonuclease Dicer recognizes and cleaves the viral dsRNA replicative intermediates into virus-derived small interfering RNAs (vsiRNAs). Subsequently, the Argonaute protein (AGO) of the RNA-induced silencing complexes (RISC) utilizes vsiRNAs to de-struct viral RNAs in the infected cells [129].

To evade immune response, ZIKV encodes viral suppressors of RNAi (VSR) to an-tagonize RNAi-mediated antiviral immunity [130]. The VSR activity of ZIKV NS2A suppressed antiviral RNAi in vitro through the inhibition of vsiRNA production [131]. ZIKV C protein interacts and antagonizes the endoribonuclease activity of host Dicers in human NSCs [132,133]. The modulation of RNAi activity through hijacking host Dicers represents a refined mechanism for ZIKV immune evasion."

  1. We have added sentences to conclude the Section 3.1 (Page 5 Line 200-201) and Section 3.2 (Page 6 Line 224-226), as follow. For section 3.5, we cannot find any other related information at present.

Section 3.1 :  "This initiates a signaling cascade, leading to the expression of type I IFNs and IFN-stimulated genes (ISGs), as elaborated below."

Section 3.2 : " This missense SNP in TLR3 caused the decreased binding capacity of dsRNA, resulting in impaired antiviral activity and an increased ZIKV viral load [88]."

  1. We have revised the sentences as follow. Missing DOIs have been added.

Page 5 Line 176-179 : "These receptors play crucial roles in triggering signaling pathways and initiating an antiviral response in the host [73]."

Page 6 Line 243-245 : "TLRs activate the myeloid differentiation primary response (MyD88), and they transmit signals via signaling molecules such as TNF receptor associated factor 3 (TRAF3) and TRAF6 [90,91]."

Page 6 Line 260-262 : "The ISGF3 is then translocated into the nucleus and transcribes ISGs, as well as the SOCS, which negatively regulate the JAK/STAT pathway by ubiquitinating JAK1 and promoting degradation through the proteosome [99]."

  1. We have corrected the typo in the manuscript, as suggested. In addition, the manuscript has been corrected by a professional English editing service to avoid any errors.

Reviewer 2 Report

This a review of the lifecycle and pathogenesis of Zika virus focusing on molecular interactions between the virus and the human host. Considering that only five of the cited references were published in 2021 or 2022, the review is not up to date and does not add new information over other reviews listed below, which include one published in Vaccines in 2021 and another one published in 2022 in International Journal of Molecular Sciences, which are also MDPI journals.

I am not an author on any of the references listed below, so there is no conflict of interest. The writing is atrocious. When I review a poorly written paper, my practice is to list 25 errors in writing as a small sample to prove the point, but the authors should be aware that the submitted manuscript contains more than 100 errors in word choice and English grammar.

Line 19, change 'the global health' to 'global health'

Line 19, change 'in majority' to 'in the majority'

Line 28, change 'host interaction' to "host-virus interaction'

Line 28, change 'provides platform' to 'provides a platform'

Lines 36-37, change 'in year 1947' to 'in 1947'

Lines 38-39, change 'are presented with no or mild symptoms' to 'have no or mild symptoms'

Line 41, change 'in South America continent' to 'in Latin America' [technically several of the countries most impacted are not in South America]

Lines 41-42, delete 'among the society'

Lines 46-47, change 'neonates of the infected pregnant women' to 'children born to mothers infected during pregnancy'

Line 47, change 'vaccine being approved' to 'vaccine approved'

Line 48, end the sentence with '[20, 21].'; start the next sentence with 'Hence, further'

Line 49, change 'of next potential' 'of a future'

Line 54, change 'ZIKV genome' to 'The ZIKV genome'

Line 66, Delete 'The protein is synthesized as monomers' because that is true for every protein and adds no information

Line 67, change 'various location' to 'various locations'

Lines 74-75, change 'that NS2A protein participates in ZIKV induced neurological damages' to 'that the NS2A protein participates in ZIKV-induced neurological damage' [three changes]

Line 84, change 'inhibition the" to 'inhibiting the'

Line 86, change 'disrupt autophagy process' to 'disrupt autophagy'

Line 95, change 'transability' to 'transmissibility'

Line 99, change 'Low virulent Asian strain, on the other hands, causes' to 'In contrast, the low virulence Asian strain causes'

Line 117, change 'When ZIKV reaches clathrin expressing surface, host cell membrane invaginates' to 'When ZIKV reaches a clathrin expressing surface, the host cell membrane invaginates'

Line 135, When did the "Yap Island outbreak' occur?

Lines 135-136, change 'majority of the ZIKV infection has been implicated with mild clinical signs characterized by' to 'the majority of cases were mild with clinical symptoms including:"

Line 137, change 'neurological implications' to 'neurological complications'

Lines 139-141, the concepts of 'incidence' and 'prevalence' are mixed in the same paragraph causing confusion

Line 190, the phrase 'while pharmacological inhibition of TLR3 [72].' is not part of any sentence

Other similar recent reviews that are collectively better than this review:

J. Estevez-Herrera et al., Zika Virus Pathogenesis: A Battle for Immune Evasion. Vaccines 2021; 9:294.

M. Guo et al. ZIKV viral proteins and their roles in virus-host interactions. Science Life China Sciences 2021; 64: 709-719.

L. J. Lee et al., Hide and seek: The interplay between Zika virus and the host immune response, Frontiers in Immunology 2021; 12:750365

C. Mwaliko et al., Zika virus pathogenesis and current therapeutic advances. Pathogens and Global Health 2021; 115:21-39.

J. Turpin et al., Apoptosis during Zika virus infection: Too soon or too late? International Journal of Molecular Sciences 2022; 23:1287.

Author Response

We sincerely thank the reviewer for valuable comments provided to improve our manuscript. Our response to the comments are as follow:

  1. We appreciate the comments provided by the reviewer, we have cited more of the related recent publications into our manuscript. In total, there are 18 recent publications from 2021 and 2022, while 56 publications are from the past 5 years.

  1. We thank the reviewer for pointing out multiple grammatical s, and we have edited accordingly. In addition, the manuscript has been corrected by a professional English editing service to avoid any errors.

Round 2

Reviewer 1 Report

The authors did respond to comment number 3 (Authors should also include the immune evasion mechanism of ZIKV capsid interaction with endoribonuclease in the immune-privileged organs in a separate section), but still, the mechanism in the immune-privileged organs has not been discussed. Authors are advised to rewrite this section with emphasis on the immune-privileged organs. Apart from this, everything looks good now.

Author Response

Author Response: We sincerely thank the reviewer for providing valuable feedback on our manuscript.

Page 10 Line 394-412: We have revised these paragraphs by adding in all details that we can find regarding the mechanism in the immune-privileged organ, as follow. If we miss out any content or references, please kindly advise so we can include in revision. Thank you!

“ZIKV C protein is also a VSR that directly interacts and antagonizes the endoribonuclease activity of host Dicers in human NSCs [132,133]. The vsiRNA production is Dicer-dependent, as evidenced by knockdown of Dicer in the RNAi pathway resulting in reduced vsiRNA and enhanced ZIKV replication in NSCs [133]. Besides, Enoxacin, a RNAi enhancer, has been shown to inhibit ZIKV-induced phenotypes associated with microcephaly by increasing RNAi in brain organoids [134], emphasizing the significant role of antiviral RNAi against ZIKV infection.

In the ZIKV-infected mouse embryonic brain, capsid-mediated Dicer inhibition disrupts the production of host microRNAs (miRNA) that are essential for neural development (i.e. let-7a, miR-9, miR-17, and miR-19a), thus causing severe defects in neurogenesis in vitro and corticogenesis in utero [132]. Whereas using a capsid-H41R mutant, ZIKV reduces its pathogenicity to cause neurologic deficits, which is due to the loss of capsid-Dicer interaction and failure to inhibit miRNA production. Accordingly, ZIKV dysregulates miRNA-mRNA interaction network that negatively impacts several biological processes such as cell cycle and neurogenesis in human NSCs [135], as well as in fetal astrocytes SVG-A cells [136]. Remarkably, the modulation of RNAi activity through hijacking host Dicers, vsiRNA and miRNA productions represents a refined mechanism for ZIKV immune evasion and favor its replication.”

References added:

  1. Zeng, J.; Dong, S.; Luo, Z.; Xie, X.; Fu, B.; Li, P.; Liu, C.; Yang, X.; Chen, Y.; Wang, X.; et al. The Zika Virus Capsid Disrupts Corticogenesis by Suppressing Dicer Activity and miRNA Biogenesis. Cell Stem Cell 2020, 27, 618-632 e619, doi:10.1016/j.stem.2020.07.012.
  2. Zeng, J.; Luo, Z.; Dong, S.; Xie, X.; Liang, X.; Yan, Y.; Liang, Q.; Zhao, Z. Functional Mapping of AGO-Associated Zika Virus-Derived Small Interfering RNAs in Neural Stem Cells. Front Cell Infect Microbiol 2021, 11, 628887, doi:10.3389/fcimb.2021.628887.
  3. Xu, Y.P.; Qiu, Y.; Zhang, B.; Chen, G.; Chen, Q.; Wang, M.; Mo, F.; Xu, J.; Wu, J.; Zhang, R.R.; et al. Zika virus infection induces RNAi-mediated antiviral immunity in human neural progenitors and brain organoids. Cell Res 2019, 29, 265-273, doi:10.1038/s41422-019-0152-9.
  4. Dang, J.W.; Tiwari, S.K.; Qin, Y.; Rana, T.M. Genome-wide Integrative Analysis of Zika-Virus-Infected Neuronal Stem Cells Reveals Roles for MicroRNAs in Cell Cycle and Stemness. Cell Rep 2019, 27, 3618-3628 e3615, doi:10.1016/j.celrep.2019.05.059.
  5. Kozak, R.A.; Majer, A.; Biondi, M.J.; Medina, S.J.; Goneau, L.W.; Sajesh, B.V.; Slota, J.A.; Zubach, V.; Severini, A.; Safronetz, D.; et al. MicroRNA and mRNA Dysregulation in Astrocytes Infected with Zika Virus. Viruses 2017, 9, doi:10.3390/v9100297.

Reviewer 2 Report

I recommended to reject this manuscript and I continue to recommend to reject it.

Author Response

We thank the reviewer for revising the manuscript.